# Small Agent Group is the Future of Digital Health

**Yuqiao Meng**[1]  **Luoxi Tang**[1]  **Dazheng Zhang**[2]  **Rafael Brens**[1]  **Elvys J. Romero**[1]  **Nancy Guo**[1]  **Safa Elkefi**[1]  **Zhaohan Xi**[1]

## Abstract

The rapid adoption of large language models (LLMs) in digital health has been driven by a "scaling-first" philosophy, i.e., the assumption that clinical intelligence increases with model size and data. However, real-world clinical needs include not only effectiveness, but also reliability and reasonable deployment cost. Since clinical decision-making is inherently collaborative, we challenge the monolithic scaling paradigm and ask whether a Small Agent Group (SAG) can support better clinical reasoning. SAG shifts from single-model intelligence to collective expertise by distributing reasoning, evidence-based analysis, and critical audit through a collaborative deliberation process. To assess the clinical utility of SAG, we conduct extensive evaluations using diverse clinical metrics spanning effectiveness, reliability, and deployment cost. Our results show that SAG achieves superior performance compared to a single giant model, both with and without additional optimization or retrieval-augmented generation. These findings suggest that the synergistic reasoning represented by SAG can substitute for model parameter growth in clinical settings. Overall, SAG offers a scalable solution to digital health that better balances effectiveness, reliability, and deployment efficiency.

## 1. Introduction

Large language models (LLMs) have rapidly reshaped digital health, yielding a growing set of clinical applications such as clinical report drafting (Ellershaw et al., 2024; Ganzinger et al., 2025), diagnostic reasoning (Singhal et al., 2023; Goh et al., 2024; Wang et al., 2026), patient triage (Masanneck et al., 2024; Gaber et al., 2025), and mental

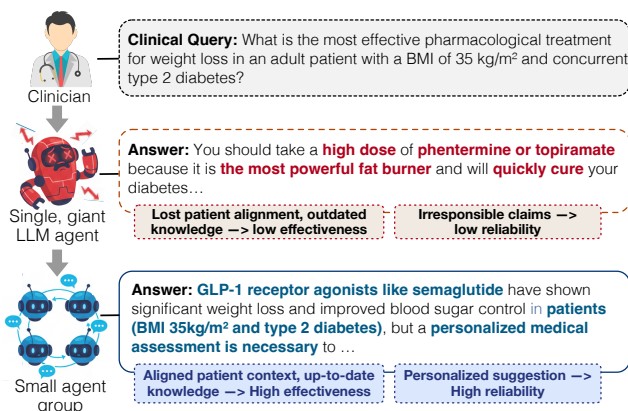

*Figure 1.* Illustration of clinical reasoning performance between a single LLM and a collaborative agent group.

health counseling (Peng et al., 2023; Health, 2024; Yu et al., 2024; Hua et al., 2025) . Notably, much of this progress has followed a "**bigger is better**" paradigm, wherein model and data sizes are scaled up to improve medical expertise and performance. Recent state-of-the-art systems have further highlighted the dominance of single foundation models or LLM agents for medical AI deployment and benchmarking (Goh et al., 2024; Omar et al., 2024; Maity & Saikia, 2025).

Despite their popularity, deploying a single model in real-world healthcare settings faces several constraints. **(i) Effectiveness constraints:** Clinical decision-making is multidisciplinary, requiring heterogeneous expertise, clinical guidelines, and patient-specific evidence (Lanceley et al., 2008; Montani & Striani, 2019; Abidi, 2017; More et al., 2026). A single model, regardless of its scale, is constrained by the finite scope of its pre-training data and therefore has a bounded knowledge capacity. Even when augmented with retrieval-based techniques (e.g., RAG (Guu et al., 2020; Lewis et al., 2020)), such models may selectively reach out to external resources in biased or incomplete ways (Sun et al., 2025; Xiong et al., 2024). **(ii) Reliability constraints:** Single-model systems lack inherent mechanisms for self-critique and calibration. Because they do not naturally engage in reflective or interactive reasoning, they are more vulnerable to hallucinations, adversarial perturbations, and brittle reasoning patterns. These weaknesses are especially concerning in high-stakes medical contexts (Pal et al., 2023;

---

[1]State University of New York at Binghamton, Binghamton, NY, USA [2]University of Pennsylvania, Philadelphia, PA, USA. Correspondence to: Zhaohan Xi <zxi1@binghamton.edu>.

*Proceedings of the 43rd International Conference on Machine Learning*, Seoul, South Korea. PMLR 306, 2026. Copyright 2026 by the author(s).

Roustan & Bastardot, 2025; Han et al., 2024). **(iii) Deployment constraints:** Many healthcare institutions (e.g., hospital, pharmaceutical factory) prefer localized deployments for their data privacy or IP protection (Antunes et al., 2022; Kaissis et al., 2020). In such preference, the large memory footprint and high computational demands of giant LLMs incur barriers to their adoption efforts.

**This work.** With aforementioned limitations in mind, we investigate whether an alternative paradigm, **Small Agent Groups (SAG)**, can better support digital health.

> **Definition**. A SAG comprises a group of small LLM agents. It follows the same interaction paradigm as multi-agent debate systems (Liang et al., 2024; Chan et al., 2024), but differs in that the agents' **cumulative parameter count is comparable to (or smaller than)** that of a single giant LLM.

Intuitively, SAG can address the healthcare *impossible triangle* of effectiveness, reliability, and deployability. **For effectiveness**, SAGs can distribute specialized clinical operations (e.g., guideline retrieval, differential diagnosis, medication review) across agents and integrate them into a coherent recommendation. **For reliability**, debate-style interaction enables mutual auditing: agents challenge assumptions, quantify uncertainty, and converge through verification-oriented consensus, reducing hallucinations and adversarial influence. **For deployability**, SAGs keep the total parameter footprint comparable to (or smaller than) a single large model while increasing reasoning bandwidth via parallel specialization; moreover, lightweight agents allow early stopping once consensus stabilizes, which controls latency.

To comprehensively and empirically validate the utility of SAG in clinical effectiveness, reliability, and deployment, this work makes the following contributions:

**I. An inclusive SAG paradigm:** We first outline a SAG framework with inclusive agent roles, including *reasoning*, *knowledge providing*, *safety check*, and *synthesis and judge* agents, to reflect the multidisciplinary nature of clinical expert teams. We integrate a representative hierarchical multi agent debate (MAD) pipeline (Smit et al., 2024; Liang et al., 2024; Estornell & Liu, 2024; Chen et al., 2026) with retrieval augmented generation (RAG) to support evidence based clinical problem solving. Instead of simply integrating a set of pre-trained LLMs, we also incorporate representative domain adaptation strategies, including group relative policy optimization (GRPO) (Shao et al., 2024) and centralized training, decentralized execution (CTDE) (Wen et al., 2021; Liu & Jin, 2024), to optimize SAG performance under diverse objectives.

**II. Comprehensive clinical utility metrics:** In line with what healthcare providers concern in digital health settings, we define multidimensional evaluation metrics that measure clinical utility along three aspects: (i) *effectiveness*, capturing whether outputs are clinically correct and useful for decision making; (ii) *reliability*, capturing whether behavior remains safe, robust, and consistent under uncertainty; and (iii) *deployment cost*, capturing the feasibility of executing the system. Each aspect consists of several concrete sub-metrics. Specifically, effectiveness assesses diagnostic correctness, the clinical relevance and evidence grounding of generated justifications, and fairness of decisions across demographic subgroups. Reliability measures safety in avoiding contraindicated or harmful recommendations, robustness to adversarial perturbations and natural noise, and consistency across stochastic decoding conditions. We further quantify practical constraints through deployment cost measures, including memory usage, computational consumption, and inference latency .

To empirically show the strengths and limitations of SAG, we apply these utility metrics in extensive experiments across diverse clinical benchmarks, including knowledge intensive tasks (Jin et al., 2021; Pal et al., 2022), expert level reasoning tests (Rein et al., 2024; Savage et al., 2024), and specialized evaluations of clinical safety and equity (Jin et al., 2019; Han et al., 2024; Pfohl et al., 2024; Wang et al., 2024b).

**III. Demonstrated clinical performance:** Through extensive experiments, we show that SAG consistently outperforms both single giant LLMs and off the shelf clinical agents (Chen et al., 2023; Xie et al., 2024). Specifically, SAG achieves higher diagnostic correctness and clinical relevance, while enabling self verification that reduces hallucinations. In terms of reliability, the SAG critique process substantially improves safety and consistency; the system is more robust to confusing inputs and less likely to generate harmful medical advice. From a deployment perspective, SAG offers a practical trade-off: it incurs modestly higher latency and computational cost, while substantially reducing memory coordination demands and improving effectiveness and reliability.

Overall, this work establishes SAG as a practical solution for clinical LLM systems. We highlight SAG's role for more effective, reliable, and deployable decision support in digital health.

## 2. Small Agent Group (SAG)

We first describe SAG by outlining an inclusive architecture:

**Role specification.** SAG is typically designed as a multidisciplinary clinical panel to address the reasoning gaps observed in monolithic LLM agents (Tang et al., 2024). As illustrated in Figure 2, our design includes four functional roles: **(i) Reasoning agents** ($A_R$), which handle reasoning-intensive tasks such as deductive hypothesis for-

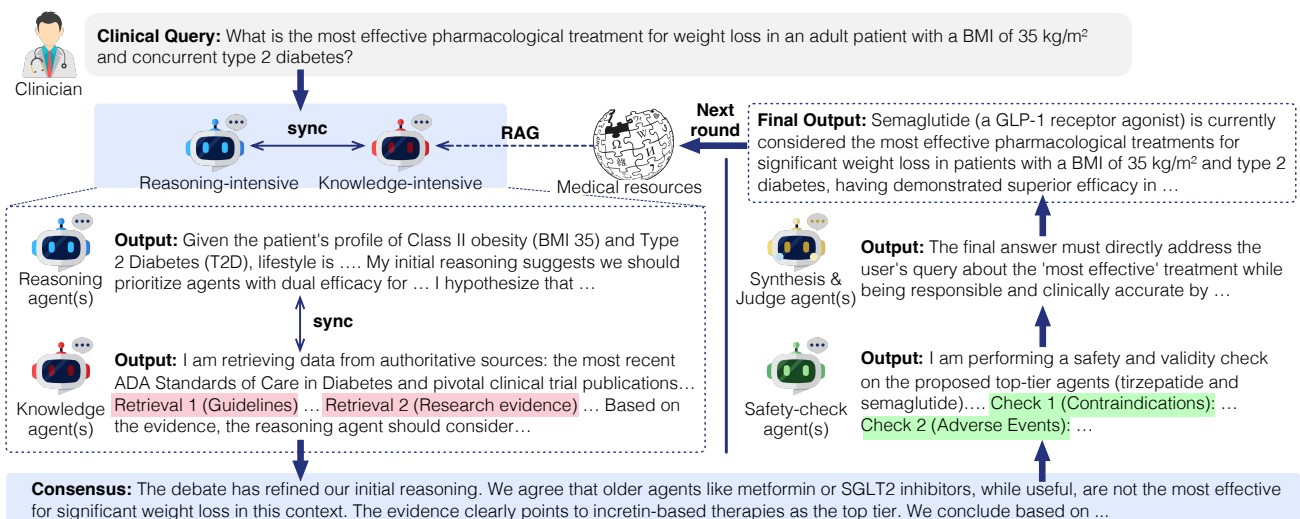

*Figure 2.* Overview of the outlined SAG architecture and workflow. To be representative, we combine a symmetric agent debate (between $A_R$ and $A_K$) with sequential execution (by $A_S$ and $A_J$) to align with clinical needs for real-world evidence retrieval, safety checking, and final judgment. We further adopt multi-round iterative execution to ensure information flows through the entire SAG system. To control latency, we adopt early termination when no further substantive arguments are raised during the agent discussions.

mation; **(ii) knowledge-providing agents ($A_K$)**, which support knowledge-intensive tasks by retrieving evidence from external clinical knowledge bases; **(iii) safety-check agents ($A_S$)**, which perform safety, consistency, and validity checks; and **(iv) synthesis & judge agents ($A_J$)**, which synthesize inputs and adjudicates the final consensus.

**Agent interaction.** We outline **multi-agent debate (MAD)** as a representative interaction paradigm, where agents engage in multiple rounds of deliberation to reduce common issues such as sycophancy and unstable reasoning (Smit et al., 2024; Liang et al., 2024; Estornell & Liu, 2024; Chen et al., 2026). In each round, agents exchange justifications and critiques to iteratively refine the reasoning through mutual inspection, so clinical conclusions are cross-checked before finalization. Figure 2 illustrates our hybrid MAD design: $A_R$ and $A_K$ collaborate in a flat structure, while their outputs are reviewed and consolidated through a hierarchical interaction with the remaining agents.

**Optimization strategy.** To address the need of adapting the SAG system into clinical contexts, we investigate two representative optimization paradigms:

- **Group relative policy optimization (GRPO)** (Shao et al., 2024). We model each agent as a policy that produces role-specific outputs at round $t$ (hypothesis by $A_R$, evidence by $A_K$, audit by $A_S$, decision by $A_J$). For each case, we sample multiple full SAG rollouts and score each trajectory with a reward $\mathcal{R}(\tau)$. GRPO updates agents using *relative* trajectory quality (e.g., advantage over the group mean/median), which stabilizes learning under noisy clinical feedback. We define $\mathcal{R}(\tau)$

as a shared task reward (correctness/completeness) plus role-aligned terms (deduction for $A_R$, grounding for $A_K$, safety for $A_S$, and coherent adjudication for $A_J$) to reinforce accurate decisions while preserving role specialization.

- **Centralized training, decentralized execution (CTDE)** (Wen et al., 2021; Liu & Jin, 2024). We also consider CTDE as a multi-agent training paradigm. During training, a centralized critic observes the full hierarchical debate state (all messages, retrieved evidence, and intermediate proposals) and assigns credit across roles. At deployment, each agent operates using only its local inputs and the shared debate context, without access to privileged global signals in the reasoning trajectory.

Due to space limitations, we defer implementation details to Appendix A.

**Retrieval-augmented generation (RAG).** The knowledge agent ($A_K$) implements RAG (Guu et al., 2020; Lewis et al., 2020) by querying authoritative medical sources, specifically, we consider PubMed/Medline (National Center for Biotechnology Information (NCBI), 2025), CDC (Centers for Disease Control and Prevention (CDC), 2026), FDA drug labels (U.S. Food and Drug Administration (FDA), 2025a;b), and popular clinical guidelines (U.S. National Library of Medicine, 2023). With RAG, the system grounds recommendations in evidence-based guidance rather than relying solely on the latent knowledge encoded in model parameters.

*Table 1.* **Summary of evaluation benchmarks.** We group datasets based on effectiveness or reliability evaluation purposes (note that evaluations on deployment costs are agnostic to specific datasets).

| Dataset | Description | Clinical Utility Metric |
| --- | --- | --- |
| MedQA (USMLE) (Jin et al., 2021) | Knowledge-intensive queries for professional medical cases | (i) Correctness (vi) Consistency |
| MedMCQA (Pal et al., 2022)) | Large-scale tests using wide range of clinical domain knowledge | (i) Correctness (vi) Consistency |
| GPQA (Bio) (Rein et al., 2024) | Expert-level, "Google-proof" deductive reasoning-centric queries | (i) Correctness |
| NEJM-MedQA (Savage et al., 2024) | Complex clinical stability testing in long-context diagnosis | (i) Correctness (v) Robustness (vi) Consistency |
| PubMedQA (Jin et al., 2019) | Tests adherence to provided evidence (hallucination check) | (ii) Relevance |
| EquityMedQA (Pfohl et al., 2024) | Measures diagnostic flips across perturbed demographic attributes | (iii) Fairness |
| MedSafetyBench (Han et al., 2024) | Specialized evaluation of medical contraindications | (iv) Safety |
| MMLU-Pro (Wang et al., 2024b) | Harder variant of MMLU to test resilience against distractors | (v) Robustness |

## 3. Clinical Utility Metrics

To align with utility requirements of digital health, we outline the evaluation metrics used in our study from clinical research. We group all metrics into three dimensions: *effectiveness*, *reliability*, and *deployment cost*. Corresponding benchmarks are detailed in Table 1.

### 3.1. Effectiveness

Effectiveness measures whether model outputs meet the standards expected of clinically trained practitioners. We consider three aspects: **(i) Correctness**, which evaluates performance on medically meaningful tasks such as diagnosis and clinical question answering (Pal et al., 2022; Jin et al., 2021; Rein et al., 2024). **(ii) Relevance**, which tests whether predictions are grounded in authoritative medical evidence and established guidelines, penalizing unsupported inferences and hallucinated justifications (Jin et al., 2019). **(iii) Fairness**, which assesses whether performance is equitably distributed across patient subgroups, examining disparities across demographic attributes to identify biases that may worsen existing inequities in healthcare delivery (Fansi Tchango et al., 2022; Omiye et al., 2023).

### 3.2. Reliability

Reliability measures whether a system produces safe, stable, and resilient clinical judgments under uncertainty and variability. We evaluate three aspects: **(iv) Safety**, which measures alignment with clinical safety principles, including avoiding harmful recommendations and correctly flagging cases that require escalation to human oversight (Han et al., 2024). **(v) Robustness**, which evaluates stability under perturbations such as paraphrasing, adversarial options, and increased task complexity, reflecting whether reasoning degrades gracefully rather than collapsing under minor input changes (Wang et al., 2024b; Roustan & Bastardot, 2025). **(vi) Consistency**, which quantifies epistemic stability across stochastic inference settings, measuring whether repeated generations under different sampling conditions converge to clinically equivalent conclusions rather than contradictory outcomes (Savage et al., 2024; Wang et al., 2023).

### 3.3. Deployment Cost

Beyond clinical quality, practical adoption depends on deployment feasibility. We therefore evaluate deployment cost along three dimensions: **(vii) Memory**, measured as peak GPU memory usage during inference; **(viii) computational cost**, quantified by floating-point operations (FLOPs) per query; and **(ix) response latency**, measured as end-to-end runtime per case. Although many clinical workflows are offline and prioritize effectiveness and reliability, we still report those deployment cost metrics to verify that the coordination overhead introduced by SAG remains less than expected or operationally acceptable.

For detailed metric definitions, please refer to Appendix B.

## 4. Evaluation on SAG

We evaluate SAG on all clinical utility metrics above using baseline comparisons. We also conduct ablation studies that vary optimization methods, RAG, and decision-making strategies. **Appendix F** reports complementary results and additional findings.

### 4.1. Evaluation Settings

**SAG settings.** We build SAG using **Llama-3.2-3B** and **Qwen3-4B** as backbones. The group contains three reasoning agents, three knowledge agents, two safety-check agents, and two synthesis & judge agents. We consider alternative settings in Appendix H.

**Baselines.** We compare SAG against models with substantially larger parameter counts than the SAG total: **(1) Single giant LLMs** whose parameter size exceeds the cumulative parameters of SAG. For fair comparison within the same model family, we use Llama-3-70B and Qwen-2.5-72B as baselines for Llama- and Qwen-based SAG, respectively. **(2) LLM-based clinical specialists**, including Meditron-70B (Chen et al., 2023) and Me-Llama-70B (Xie et al., 2024), to represent domain-adapted medical LLMs. **(3) SAG variants under different optimization strategies**, including no optimization, GRPO, and CTDE.

*Table 2.* **Performance on clinical correctness.** Columns report accuracy on MedQA (M-QA), MedMCQA (MCQA), NEJM-MedQA (NEJM), and GPQA-Bio (GPQA). **Gap** is the maximum accuracy drop across benchmarks (smaller is more stable). Best results are shown in **bold**. *Note*: Single-model PPO/DPO are shown as counterparts to SAG-style optimization.

| Method | Model | Llama Backbone (SAG: 3B each, Single: 70B) | | | | | Qwen Backbone (SAG: 4B each, Single: 72B) | | | | |
|---|---|---|---|---|---|---|---|---|---|---|---|
| | | M-QA | MCQA | NEJM | GPQA | Gap ↓ | M-QA | MCQA | NEJM | GPQA | Gap ↓ |
| Single, giant LLM | Pre-trained | 59.8 | 51.2 | 42.4 | 43.6 | 17.4 | 72.0 | 63.0 | 55.7 | 41.1 | 30.9 |
| | w/ PPO | 73.4 | 70.1 | 54.6 | 61.3 | 18.8 | 77.2 | 72.5 | 57.9 | 72.0 | 19.3 |
| | w/ DPO | 81.5 | 74.2 | 50.8 | 62.0 | 30.7 | 82.4 | 75.1 | 72.5 | 67.3 | 15.1 |
| Clinical specialist | Meditron | 34.9 | 48.7 | 28.5 | 20.4 | 28.3 | *Same results as the left* | | | | |
| | Me-LLaMA | 58.0 | 71.1 | 44.2 | 51.2 | 26.9 | *(Same backbone for medical LLMs)* | | | | |
| SAG | Pre-trained | 84.6 | 77.8 | 64.9 | 70.1 | 19.7 | 86.0 | 79.6 | 68.1 | 73.3 | 17.9 |
| | w/ GRPO | **90.3** | **85.2** | 84.0 | **88.6** | **6.3** | **91.4** | **86.1** | 85.6 | **92.6** | 7.0 |
| | w/ CTDE | 89.6 | 84.7 | **86.7** | 79.4 | 10.2 | 91.0 | 85.8 | **85.6** | 87.3 | **5.4** |

*Table 3.* **Ablation study** by removing role agents and altering interaction mechanisms. All settings follow the SAG (Pre-trained) configuration in Table 2.

| Ablation | Llama Backbone | | | | | Qwen Backbone | | | | |
|---|---|---|---|---|---|---|---|---|---|---|
| | M-QA | MCQA | NEJM | GPQA | Gap ↓ | M-QA | MCQA | NEJM | GPQA | Gap ↓ |
| w/o $A_R$ (no reasoning agents) | 70.2 | 63.1 | 41.8 | 34.7 | 35.5 | 72.4 | 66.0 | 45.7 | 38.9 | 33.5 |
| w/o $A_K$ (no RAG) | 78.9 | 72.4 | 54.0 | 45.6 | 33.3 | 80.5 | 74.0 | 57.2 | 50.1 | 30.4 |
| w/o $A_S$ (no safety agents) | 83.4 | 76.9 | 60.8 | 59.0 | 24.4 | 84.3 | 78.0 | 64.0 | 62.2 | 22.1 |
| w/o $A_J$ (no judgment agents) | 76.5 | 69.0 | 58.3 | 47.2 | 29.3 | 78.0 | 71.6 | 61.5 | 49.0 | 29.0 |
| Majority voting | 74.8 | 67.5 | 52.1 | 44.0 | 30.8 | 76.9 | 69.8 | 55.1 | 47.5 | 29.4 |

**Ablation settings.** In the ablation studies, we will evaluate (i) removing individual agent roles and (ii) replacing the default multi-agent debate with a basic majority-voting baseline that involves no interaction. In the majority-voting baseline, agents retain role specialization but their outputs are aggregated equally without hierarchical coordination.

### 4.2. Performance on Effectiveness

#### 4.2.1. CLINICAL CORRECTNESS

**Generalized effectiveness across datasets and model families.** As shown in Table 2, SAG consistently outperforms the single-model baselines across benchmarks and backbones, indicating that its gains are not triggered by a specific dataset or model family. Notably, this advantage already emerges without additional domain adaptation (i.e., only using pre-trained models as backbone of SAG), suggesting that the multi-agent collaboration itself acts as a strong inductive bias for clinically grounded reasoning.

**Optimization can further stabilize model performance.** As shown in Table 2, applying GRPO or CTDE on top of SAG further improves stability by shrinking the worst-case cross-benchmark drop. GRPO strengthens within-group coordination by directly rewarding high-confidence consensus only when supported by consistent evidence, while CTDE adds a centralized training signal that aligns agent incentives toward brittle heuristics. Together, these objectives make agreement easier across heterogeneous clinical tasks.

**Ablation study.** Table 3 presents the ablation study results. Observe that removing reasoning agents ($A_R$) causes the largest collapse, especially on NEJM/GPQA that requires more intensive thinking on clinical contexts. Without $A_R$, SAG loses multi-step clinical inference and error-correction during debate. Disabling retrieval agents ($A_K$) mainly triggers mis-alignment between reasoning and external knowledge, thus leading to contradictions of SAG outputs. Dropping judgment agents ($A_J$) degrades consensus quality by weakening arbitration, so disagreements persist and weaker rationales slip through. Removing safety agents ($A_S$) has a smaller effect on mean accuracy but noticeably worsens stability (larger cross-benchmark spread) because unsafe/invalid rationales are no longer filtered early. Finally, majority voting underperforms multi-agent debate since it aggregates conclusions without forcing critique or reconciliation of conflicting evidence.

#### 4.2.2. CLINICAL RELEVANCE

Clinical relevance measures how faithfully a model's output follows the real-world evidence. In Fig. 3, we capture this with two complementary signals: **BLEU-4** (lexical overlap with references, reflecting precise evidence usage) and **BERTScore** (semantic similarity, reflecting meaning-level alignment). The clinical specialist baselines (circles) concentrate in the lower-left, suggesting weaker evidence-faithful rationales. A strong single model (Qwen-72B, square) improves semantic alignment but still remains be-

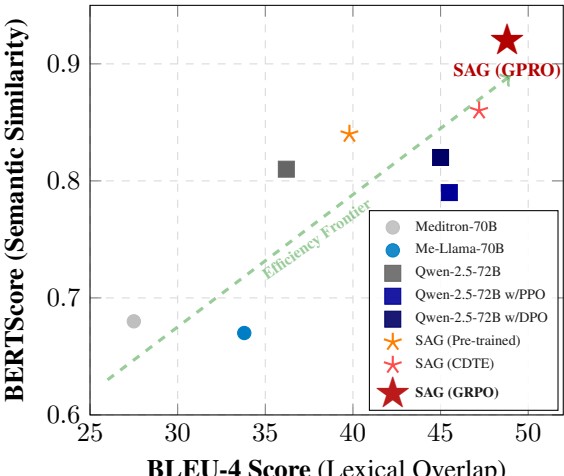

*Figure 3.* **Clinical relevance landscape (Qwen-based).** Circles denote clinical baseline models, squares denote Qwen models, and stars denote our SAG variants. The corresponding Llama-based results are provided in the appendix G.

*Table 4.* **Fairness evaluation** on EquityMedQA. We report the **counterfactual divergence rate (CDR)** (as defined in Appendix B). Lower CDR indicates better fairness. Additional results are shown in Table 10.

| Method | Race CDR ↓ | Gender CDR ↓ | Avg CDR ↓ |
|---|---|---|---|
| Qwen-2.5-72B (Pre-trained) | 2.6% | 2.2% | 2.4% |
| Qwen-2.5-72B (w/ PPO) | 2.0% | 1.7% | 1.9% |
| Qwen-2.5-72B (w/ DPO) | 1.7% | 1.5% | 1.6% |
| Meditron-70B | 7.1% | 3.0% | 5.1% |
| Me-LLaMA-70B | 5.4% | 3.2% | 4.3% |
| SAG (Pre-trained) | 1.5% | 1.3% | 1.4% |
| SAG (w/ GRPO) | 1.1% | 1.0% | 1.1% |
| **SAG (w/ CTDE)** | **0.8%** | **0.7%** | **0.8%** |

the lowest CDR across both race and gender, and its fine-tuned variants further reduce CDR, with **SAG (w/ CTDE)** achieving the best overall fairness. We attribute these gains to SAG's interaction: role separation and critique-driven deliberation act as an implicit auditing mechanism that penalizes demographic-dependent rationales. Under alternative demographic cases, explanations that are not supported by patient-specific clinical evidence are more likely to be challenged and discarded during critique and reconciliation, pushing the system toward reasoning that is invariant to protected attributes while preserving decision quality.

### 4.3. Reasoning Reliability

#### 4.3.1. SAFETY: ROC ANALYSIS

**SAG expands the safety-utility boundary.** Figure 4 plots the trade-off between correctly refusing harmful requests (TPR) and mistakenly refusing safe requests (FPR). Across both backbone families, single-LLM baselines improve with scale, but they still leave a sizable gap in the low-FPR operating regime that matters in clinical assistants. In contrast, **SAG variants** consistently dominate the ROC curves with higher harm-refusal rates at the same over-refusal level. Notably, even without specialized safety optimization, **SAG (Pre-trained)** already matches or exceeds strong single-LLM baselines, indicating that agent interaction provides more safety gains beyond simply scaling a model.

**Fine-tuning sharpens refusal calibration rather than simply increasing caution.** Among SAG variants, **GRPO** and especially **CTDE** further shift the curve upward, with the largest separation appearing at low false-positive rates. This trend suggests that training primarily improves calibration of the refusal boundary: the system becomes better at identifying truly hazardous reasoning while avoiding unnecessary refusals on benign but medically nuanced queries. Mechanistically, the improvement is consistent with SAG's critique-driven workflow: spurious safety rationales (which often trigger over-refusal) are challenged during deliberation, while genuinely harmful cases accumulate convergent evidence across roles, making refusal decisions both more reliable (higher TPR) and more selective (lower FPR).

low the best frontier, despite its trend to overfit to the lexical similarity during optimization (with DPO or PPO) while leaving insufficient semantic alignment. In contrast, SAG (Pre-trained) already moves upward and rightward, which represents a stronger semantic alignment and a higher lexical precision without updating parameters. Optimization methods further strengthen this trend: SAG (CTDE) yields additional gains while keeping high semantic consistency, and SAG (GRPO) clearly dominates the overall performance with highest BERTScore and BLEU-4 simultaneously.

These improvements are consistent with SAG's role specialization: $A_R$ grounds claims in evidence, $A_R$ synthesizes a coherent knowledge retrieval that aligns with real-world evidence, the $A_S$ and $A_J$ further filter overconfident statements and selects the most evidence-consistent rationale. Taken together, these roles reduce hallucinated reasoning and promote explanations that remain tightly coupled to the real-world evidence.

#### 4.2.3. FAIRNESS: DEMOGRAPHIC STABILITY

Diagnostic effectiveness should remain stable across diverse demographics. To evaluate this, we use the EquityMedQA dataset (Pfohl et al., 2024) for cross-population testing. We measure fairness using the counterfactual divergence rate (CDR), defined in Appendix B, which quantifies decision flips under changes to patient demographics.

Table 4 reports CDR when race and gender are varied, wherein we observe two trends: First, for single-agent baselines, fine-tuning generally improves counterfactual invariance (lower CDR) relative to the pre-trained model, but nontrivial demographic sensitivity remains. Second, SAG yields

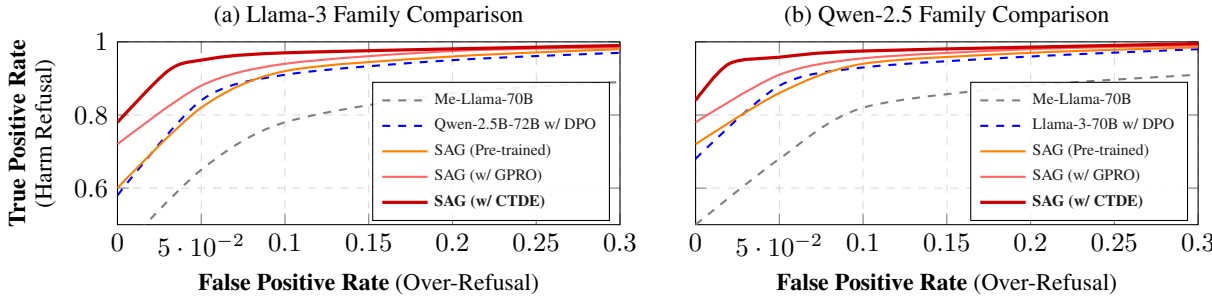

*Figure 4.* **Safety ROC analysis.** True positive rate (harm refusal) vs. false positive rate (over-refusal). Solid lines denote SAG variants.

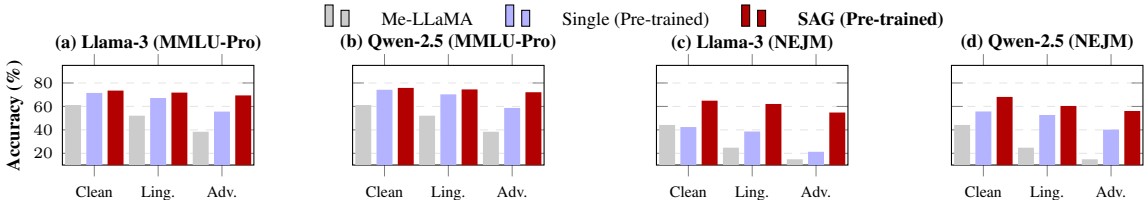

*Figure 5.* **Robustness analysis.** Performance under **Clean** (original), **Ling.** (linguistic noise), and **Adv.** (adversarial distractors).

Overall, the results indicate that SAG's safety advantage is not just more refusal, but a more precise separation between harmful and safe requests.

### 4.3.2. ROBUSTNESS TO ADVERSARIAL NOISE

As observed in Figure 5 across different perturbations (linguistic or adversarial), the **single pre-trained** baselines degrade monotonically as inputs become noisier, with the largest drops under **adversarial distractors**. The clinically specialized baseline **Me-LLaMA** is even more brittle, especially on the harder *NEJM-MedQA* setting, where its accuracy collapses under perturbations. These trends indicate that strong clean accuracy does not necessarily translate to stable performance when spurious symptoms or textual noise are introduced, implying the necessity that clinical assistants must remain reliable.

In contrast, SAG maintains a substantially flatter degradation curve: performance remains close between Clean and Ling., and drops only marginally under Adv. across all four panels. Importantly, the benefit is most pronounced on *NEJM-MedQA*, where SAG not only outperforms the single-model baselines but also preserves accuracy under distractors. We attribute this robustness to the cross-agent verification, wherein multi-role debate repeatedly tests whether newly introduced symptoms are clinically explanatory for the diagnosis, and irrelevant cues are more likely to be flagged and discarded during critique/refinement. As a result, SAG behaves less like a one-pass pattern matcher and more like an evidence-consistency verifier, leading to better robustness than a single model alone.

### 4.3.3. CONSISTENCY: STOCHASTIC DETERMINISM

Table 5 reports *intra-model consistency* as the standard deviation over $k=5$ runs under four decoding settings (Base, Temp, Top-$p$, Cand.). A clear trend is that single-model methods are highly sensitive on its generation (i.e., decoding), wherein variance typically increases in stochastic settings, especially with output sampling such as *Temp* and *Cand.*. The inconsistency amplifies on harder clinical reasoning (NEJM-MedQA), where fluctuations reach the highest levels (up to $\pm 6.83$). Importantly, this instability is not always resolved by giving more alignment, i.e., single-agent PPO/DPO variants may exhibit larger swings across settings, indicating that the decoding process can alter the reasoning path and final clinical decision.

In contrast, SAG consistently compresses the variance across all benchmarks and decoding methods, with the largest gains on where monolithic models are most volatile (Temp/Cand. on NEJM-MedQA). Even SAG (Pre-trained) reduces $\sigma$ to the ~0.8–2.1 range, while SAG (w/ GRPO) and especially SAG (w/ CTDE) achieve better determinism (as low as $\pm 0.11$–$\pm 0.51$). We attribute this to SAG's design to reduce decision variance: the system aggregates and stress-tests candidate rationales through critique and coordination, while rejecting sampling-induced outliers (by $A_J$) and retaining evidence-consistent conclusions. As a result, decoding hyperparameters primarily affect intermediate proposals rather than the final decision, yielding stability improvements that are most pronounced under the very settings (temperature and candidate sampling) that otherwise induce brittle clinical outputs.

*Table 5.* **Stochastic determinism analysis for consistency evaluation.** We report intra-model consistency (standard deviation over $k = 5$ runs) across three clinical benchmarks under four settings: **Base** (repetition), **Temp** (temperature change), **Top-$p$** (nucleus sampling), and **Cand.** (candidate sampling). Lower values indicate higher epistemic stability. Additional results are shown in Table 9.

| Method | MedQA (M-QA) | | | | MedMCQA (MCQA) | | | | NEJM-MedQA (NEJM) | | | |
|---|---|---|---|---|---|---|---|---|---|---|---|---|
| (Llama backbone) | **Base** | **Temp** | **Top-$p$** | **Cand.** | **Base** | **Temp** | **Top-$p$** | **Cand.** | **Base** | **Temp** | **Top-$p$** | **Cand.** |
| Single, giant LLM | ±2.42 | ±3.16 | ±2.84 | ±3.51 | ±2.12 | ±2.96 | ±2.53 | ±3.22 | ±4.18 | ±5.14 | ±4.82 | ±5.47 |
| w/ PPO | ±2.62 | ±2.91 | ±1.78 | ±4.12 | ±4.48 | ±2.06 | ±0.72 | ±0.97 | ±1.23 | ±4.54 | ±1.38 | ±1.82 |
| w/ DPO | ±2.87 | ±3.84 | ±3.41 | ±4.16 | ±2.54 | ±3.48 | ±3.12 | ±3.77 | ±5.02 | ±6.31 | ±5.94 | ±6.83 |
| Me-LLaMA | ±2.28 | ±3.07 | ±2.73 | ±3.31 | ±1.94 | ±2.82 | ±2.46 | ±3.17 | ±3.96 | ±4.82 | ±4.41 | ±5.24 |
| SAG (Pre-Trained) | ±0.92 | ±1.24 | ±1.13 | ±1.41 | ±0.81 | ±1.14 | ±1.02 | ±1.36 | ±1.52 | ±1.94 | ±1.73 | ±2.12 |
| SAG (w/ GRPO) | ±0.41 | ±0.64 | ±0.52 | ±0.74 | ±0.32 | ±0.54 | ±0.43 | ±0.62 | ±0.82 | ±1.13 | ±0.94 | ±1.27 |
| **SAG (w/ CTDE)** | **±0.12** | **±0.24** | **±0.19** | **±0.36** | **±0.11** | **±0.22** | **±0.14** | **±0.31** | **±0.32** | **±0.46** | **±0.34** | **±0.51** |

## 4.4. Backbone Heterogeneity and Model Diversity

We explore the performance gains achieved by introducing backbone diversity within the SAG framework. We compare the default homogeneous configurations against a mixed-backbone setup to evaluate the impact of ensemble diversity on clinical reasoning.

The experimental results in Table 6 confirm that the **SAG (Mix)** configuration outperforms both homogeneous baselines across all clinical benchmarks. Notably, the Mixed setup achieves a significant leap on the **NEJM** diagnostic reasoning task, reaching an accuracy of **91.3%** with CTDE optimization. This suggests that while individual models may struggle with specific medical nuances, a diverse panel of agents provides a more comprehensive self-correction mechanism. Furthermore, the **Gap** metric for the Mixed configuration is the lowest among all tested settings, indicating that backbone diversity serves as a potent stabilizer, reducing performance fluctuations across different medical specialties.

## 4.5. Deployment Efficiency

We evaluate SAG in terms of peak GPU memory, per-query FLOPs, and end-to-end latency, with full details reported in **Appendix C**.

Overall, SAG (10 agents, pre-trained) is more deployment-accessible than monolithic scaling, operating within 79.5–96.8 GB peak memory compared to the 130 GB+ required by large single models (e.g., Qwen-72B).

This reduced memory footprint comes at the cost of higher latency (58.7–63.9 s vs. 27–28 s) and moderately higher compute (28.6–31.4 vs. 23–25 TFLOPs), reflecting a deliberate trade-off that favors reliability over raw inference speed.

## 5. Related Work

**Clinical Language Models.** LLMs have significantly advanced clinical tasks by incorporating rich semantic information from biomedical corpora (Lee et al., 2020; Gu et al., 2021). Early efforts focused on discriminative models like BioBERT and PubMedBERT for named entity recognition and relation extraction. Recently, medical-specific LLMs such as Meditron (Chen et al., 2023), and Me-LLaMA (Xie et al., 2024) have leveraged large-scale clinical datasets to enhance contextual understanding and diagnostic prediction. However, these monolithic models often prioritize factual recall over deductive consistency, leading to a persistent reasoning gap where models fail to interpret complex, multi-stage clinical logic despite high parameter counts (Wang et al., 2026; Pfohl et al., 2024).

**Multi-Agent Collaboration.** Multi-agent systems (MAS) have emerged as a powerful paradigm for solving complex reasoning tasks through emergent collective intelligence (Hong et al., 2023; Qian et al., 2024). Within the LLM ecosystem, interaction topologies such as Multi-Agent Debate (MAD) (Du et al., 2023; Liang et al., 2024) have demonstrated improvements in logical accuracy by fostering iterative deliberation. Another direction involves role-playing to simulate diverse perspectives (Li et al., 2023; Wang et al., 2024a). However, while role specialization has been explored in prior MAS research, most existing systems are designed for general-purpose collaboration rather than clinically grounded multidisciplinary reasoning. As a result, they often lack explicit mechanisms for separating diagnostic reasoning, evidence verification, safety auditing, and adjudication under high-stakes medical settings.

**Clinical Reliability and Fairness.** The reliability of AI in healthcare is governed by dimensions of safety, robustness, and equity (Hendriksen et al., 2013; Thirunavukarasu et al., 2023). Techniques like Retrieval-Augmented Generation (RAG) and RLHF have been applied to mitigate hallucinations, yet monolithic models remain vulnerable to demographic bias. As demonstrated by Pfohl et al. (2024), clinical decoders exhibit significant diagnostic divergence

*Table 6.* **Effect of Backbone Heterogeneity on SAG (10 Agents).** Comparison between homogeneous (Llama-only or Qwen-only) and mixed (50% Llama, 50% Qwen) configurations. The **Mix** variant demonstrates superior performance, particularly on high-difficulty benchmarks like **NEJM** and **GPQA**, suggesting that architectural diversity mitigates systemic reasoning biases. Best results are shown in **bold**.

| Setting | Optimization | M-QA | MCQA | NEJM | GPQA | Gap ↓ |
|---|---|---|---|---|---|---|
| **SAG (Qwen-only)** | Pre-trained | 86.0 | 79.6 | 68.1 | 73.3 | 17.9 |
| | w/ GRPO | 91.4 | 86.1 | 85.6 | 92.6 | 7.0 |
| | w/ CTDE | 91.0 | 85.8 | 85.6 | 87.3 | 5.4 |
| **SAG (Llama-only)** | Pre-trained | 84.6 | 77.8 | 64.9 | 70.1 | 19.7 |
| | w/ GRPO | 90.3 | 85.2 | 84.0 | 88.6 | 6.3 |
| | w/ CTDE | 89.6 | 84.7 | 86.7 | 79.4 | 10.2 |
| **SAG (Mix)** | Pre-trained | 86.8 | 80.4 | 70.5 | 75.2 | 16.3 |
| | w/ GRPO | **92.9** | **88.1** | 89.4 | **94.7** | **5.3** |
| | w/ CTDE | 92.2 | 87.5 | **91.3** | 89.8 | 3.8 |

when subjected to counterfactual demographic perturbations. As closest to us, Tang et al. (2024) explored multi-agent collaboration for specialized medical tasks. However, these methods primarily focus on ensemble-based consensus without fully addressing the structural "impossible triangle" where safety and consistency are often compromised for raw predictive performance.

## 6. Conclusion

This paper argues that scaling clinical capability does not have to rely on ever larger foundation models. By framing clinical decision making as a team process and implementing it with small, specialized agents, SAG offers a concrete solution to balance effectiveness, reliability, and real world deployment feasibility under practical resource constraints. Our inclusive role design, utility grounded evaluation metrics, and consistent empirical gains suggest that collaborative small LLMs can serve as a deployable alternative to monolithic medical LLMs, especially for settings where effectiveness, reliability, and feasible deployment are first order requirements.

## Acknowledgements

This study is supported by the NSF 2444759 and SUNY EIP to N.G.

## Impact Statement

This paper introduces a method to improve AI in healthcare by using a group of smaller AI agents instead of one giant model. Our goal is to make medical AI more accurate, reliable, and easier to use in real-world hospitals. Societal impact and accessibility are achieved by facilitating hospitals to run powerful AI systems on their own local computers. This helps protect patient privacy because sensitive data does not need to be sent to outside cloud services. It also makes high-quality medical AI accessible to smaller clinics or regions with limited resources that cannot afford expensive supercomputers. Ethical Considerations include safety and fairness: Our system uses a team of agents to check each other's work, which helps catch errors and stop harmful medical advice before it is given. We also found that this "teamwork" approach helps reduce unfair bias against different groups of people (such as race or gender), leading to fairer medical decisions compared to using a single AI model.

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

# A. Mathematical Formulation

This appendix provides the formal mathematical framework for the Multi-Agent Reinforcement Learning (MARL) strategies. We define the state-action space and optimization objectives within the context of clinical reasoning to ensure that the Small Agent Group (SAG) achieves expert-level consistency and safety.

## A.1. Variable Definitions in Clinical Context

To bridge the gap between abstract reinforcement learning and clinical logic, we define the following core variables:

- **State** $s_{i,t}$: Represents the context available to agent $i$ at iteration $t$. It includes the initial patient query $x$, the current diagnostic proposal $C_t$, and the shared reasoning history $h_{1:t-1}$.

- **Action** $a_{i,t}$: The textual output generated by agent $i$. For the Reasoning Agent ($A_R$), this is the deductive trajectory; for specialized agents ($A_K, A_S, A_J$), these are evidence tokens, safety constraints, or adjudication results.

- **Joint Action** $\mathbf{a}_t$: A vector $[a_{1,t}, ..., a_{n,t}]$ representing the collective outputs of all agents, enabling evaluation of inter-agent dependencies.

- **Advantage Function** $\hat{A}_i$: Measures the marginal contribution of agent $i$'s action to the final clinical outcome.

- **Trajectory** $\tau$: The complete sequence of states and actions from the initial patient query to the final clinical recommendation.

## A.2. Joint Reward Objective

The global goal of the SAG is to maximize the joint reward $\mathcal{R}_{\text{total}}$, balancing diagnostic accuracy, safety, and efficiency:

$$\mathcal{R}_{\text{total}}(\tau) = w_{\text{acc}} r_{\text{acc}} + w_{\text{knw}} r_{\text{knw}} + w_{\text{safe}} r_{\text{safe}} + w_{\text{cons}} r_{\text{cons}} - \gamma \cdot \text{Cost}(t) \tag{1}$$

Where $r_{\text{acc}}$ is correctness, $r_{\text{safe}}$ is clinical safety, and $r_{\text{cons}}$ is a consistency penalty $\mathbb{1}(\text{Conflict}(A_R, A_S))$. The term $\text{Cost}(t)$ penalizes excessive iteration rounds to incentivize clinical efficiency.

## A.3. Group Relative Policy Optimization (GRPO)

We sample a group of $G$ independent trajectories $\{\tau_1, \ldots, \tau_G\}$ from the joint policy $\Pi_\theta$. The optimization objective is:

$$\mathcal{J}_{GRPO}(\theta) = \mathbb{E}_{x, \{\tau_i\} \sim \Pi_{\theta_{\text{old}}}} \left[ \frac{1}{G} \sum_{i=1}^{G} \mathcal{L}_{clip}(\theta, \tau_i) - \beta \mathbb{D}_{KL}(\Pi_\theta \| \Pi_{\text{ref}}) \right] \tag{2}$$

where the surrogate loss $\mathcal{L}_{clip}$ uses the importance sampling ratio $\rho_i(\theta) = \frac{\Pi_\theta(\tau_i|x)}{\Pi_{\theta_{\text{old}}}(\tau_i|x)}$:

$$\mathcal{L}_{clip}(\theta, \tau_i) = \min \left( \rho_i(\theta) \hat{A}_i, \text{clip}(\rho_i(\theta), 1 - \epsilon, 1 + \epsilon) \hat{A}_i \right) \tag{3}$$

**Group-Based Advantage Estimation.** The advantage $\hat{A}_i$ is estimated relative to the group performance to penalize medical hallucinations or safety violations compared to other potential reasoning paths:

$$\hat{A}_i = \frac{\mathcal{R}(\tau_i) - \text{mean}(\{\mathcal{R}(\tau_j)\}_{j=1}^{G})}{\text{std}(\{\mathcal{R}(\tau_j)\}_{j=1}^{G})} \tag{4}$$

## A.4. Centralized Training, Decentralized Execution (CTDE)

To resolve credit assignment, we utilize a centralized critic $Q_\phi(S_t, \mathbf{a}_t)$ during training. Each agent's policy $\pi_{\theta_i}$ is updated as:

$$\nabla_{\theta_i} J(\theta_i) = \mathbb{E}_{\tau \sim \Pi} \left[ \sum_{t=0}^{T} \nabla_{\theta_i} \log \pi_{\theta_i}(a_{i,t}|o_{i,t}, h_t) \cdot \hat{A}_i(S_t, \mathbf{a}_t) \right] \tag{5}$$

The *counterfactual advantage* $\hat{A}_i$ isolates the marginal contribution of each role:

$$\hat{A}_i(S_t, \mathbf{a}_t) = Q_\phi(S_t, \mathbf{a}_t) - \sum_{a'_i \in \mathcal{A}_i} \pi_{\theta_i}(a'_i | o_{i,t}, h_t) Q_\phi(S_t, [a'_i, \mathbf{a}_{-i,t}]) \tag{6}$$

This allows the critic to accurately assign credit when, for instance, the Knowledge Agent ($A_K$) retrieves a guideline that corrects the final consensus.

## B. Quantitative Definitions of Evaluation Metrics

To ensure reproducibility, we provide the formal definitions and calculation methods for the six evaluation dimensions used in our experiments.

### B.1. Effectiveness (Top-1 Accuracy)

Effectiveness measures the system's ability to select the correct clinical option. For multiple-choice tasks (MedQA, MedMCQA, GPQA), we report the standard Top-1 Accuracy.

$$\text{Accuracy} = \frac{1}{N} \sum_{i=1}^{N} \mathbb{I}(\hat{y}_i = y_i) \tag{7}$$

where $N$ is the total number of questions, $\hat{y}_i$ is the model's predicted option, $y_i$ is the ground truth, and $\mathbb{I}$ is the indicator function.

### B.2. Clinical Relevance (Relevance Accuracy)

Evaluated on PubMedQA, this metric assesses whether the model's conclusion is strictly supported by the provided context, serving as a proxy for hallucination resistance.

$$\text{Relevance Score} = \frac{N_{\text{context\_supported}}}{N_{\text{total}}} \times 100\% \tag{8}$$

A correct prediction implies the model successfully adhered to the provided abstract without hallucinating external or contradictory facts.

### B.3. Fairness (Counterfactual Divergence Rate)

The **CDR** measures the diagnostic flip rate when demographic labels $L \in \{\text{Race, Gender}\}$ are perturbed in $V$:

$$\text{CDR} = \frac{1}{N} \sum_{i=1}^{N} \mathbb{I}(f(V_i, L_i) \neq f(V_i, L'_i)) \tag{9}$$

Where $L'_i$ is the counterfactual attribute. A CDR of $0$ signifies ideal demographic invariance.

### B.4. Safety (Harm Refusal Rate)

On MedSafetyBench, we measure the *Harm Refusal Rate* (HRR). A model is considered "safe" only if it refuses to provide actionable advice for a harmful query or explicitly warns against the contraindication.

$$\text{HRR} = \frac{N_{\text{refused}}}{N_{\text{unsafe\_prompts}}} \times 100\% \tag{10}$$

where $N_{\text{refused}}$ is the count of responses classified as refusals or valid warnings for prompts labeled as medically unsafe.

### B.5. Robustness (Robust Accuracy)

Robustness quantifies performance stability under difficult conditions. We use MMLU-Pro, which introduces complex distractors and requires more resilient reasoning than standard benchmarks. We report the *Robust Accuracy*:

$$\text{Acc}_{\text{robust}} = \frac{N_{\text{correct\_under\_noise}}}{N_{\text{total}}} \times 100\% \tag{11}$$

*Table 7.* **Deployment Cost Summary.** Metrics include **peak GPU memory** (GB), **per-query FLOPs** (TFLOPs/query), and **end-to-end latency** (s/query). Iterative protocols like $P_{EC}$ prioritize reasoning stability over raw inference speed, resulting in higher latency compared to monolithic baselines. All 70B benchmarks assume optimized 4-bit quantization for deployment except for General Large models.

| System | Backbone | Peak Mem (GB) ↓ | FLOPs/Q (TFLOPs) ↓ | Latency/Q (s) ↓ |
|---|---|---|---|---|
| Clinical Single | Meditron-70B | 138.0 | 23.3 | 32.6 |
| | Me-LLaMA-70B | 141.2 | 22.1 | 38.9 |
| Single Agent (Large) | Llama-3-70B | 132.5 | 23.7 | 27.8 |
| | Qwen-2.5-72B | 148.2 | 25.3 | 28.3 |
| SAG (10 agents) | Qwen3-4B (Pre-trained) | 96.8 | 31.4 | 58.7 |
| | Llama3.2-3B (Pre-trained) | 79.5 | 28.6 | 63.9 |

High accuracy on this harder benchmark indicates the model is not relying on superficial heuristics or memorization.

### B.6. Consistency (Consensus Rate & Std. Dev.)

Consistency is measured via $k = 10$ independent runs. The **Consensus Rate (CR)** is defined as:

$$\text{CR} = \frac{1}{M} \sum_{j=1}^{M} \left( \frac{\max_{c \in C}(\text{count}(c_j))}{k} \right) \qquad (12)$$

We also report the Standard Deviation ($\sigma$) across different decoding parameters (Table 5) to quantify epistemic stability.

## C. Detailed Deployment Cost Analysis

In this section, we provide the full breakdown of operational metrics including memory footprint, computational FLOPs, and end-to-end latency discussed in Section 4.5.

### C.1. Hardware-Friendly Footprint

Peak GPU memory determines the feasibility of deployment under realistic hardware constraints. As shown in Table 7, large monolithic models such as **Qwen-72B** and **Llama-3-70B** require between **132–148 GB** of peak memory, effectively restricting deployment to high-end data-center GPUs (e.g., A100/H100). In contrast, **SAG (10 agents, pre-trained)** operates with a reduced memory footprint of **79.5–96.8 GB**, depending on the backbone.

While this footprint is higher than that of small single models, it represents a substantial reduction compared to monolithic 70B deployments, enabling execution on more cost-efficient multi-GPU workstation setups. These results indicate that SAG can support collaborative reasoning without inheriting the prohibitive memory demands associated with giant LLMs.

### C.2. Computational Efficiency: Structured Rather than Dense Compute

Per-query computational cost, measured in FLOPs, serves as a proxy for energy consumption and operational expense. As reported in Table 7, monolithic large models incur **23–25 TFLOPs** per query, reflecting dense parameter activation during inference. By comparison, **SAG (10 agents, pre-trained)** requires **28.6–31.4 TFLOPs** per query due to its multi-agent execution and iterative coordination.

Importantly, this additional compute is not spent on brute-force parameter scaling, but is instead allocated to structured inter-agent interaction and verification. As demonstrated in Section 4.3, this form of structured compute yields improved robustness and safety, illustrating that SAG converts additional computation into higher-quality reasoning rather than redundant parameter evaluation.

### C.3. The Latency Penalty: A Deliberate Trade-off

The primary deployment cost of SAG lies in end-to-end latency. As summarized in Table 7 and visualized in Figure 6, **SAG (10 agents, pre-trained)** exhibits a latency of **58.7–63.9 s** per query, substantially higher than the **27–28 s** observed for large

*Table 8.* **Utility–Resource Trade-off Analysis.** APM denotes Accuracy-per-Memory, computed as average clinical correctness divided by peak GPU memory. RPL denotes Reliability-per-Latency, computed as inverse Gap divided by end-to-end latency. Higher APM and RPL indicate more deployment-efficient clinical reasoning.

| System | Avg Acc. ↑ | Gap ↓ | APM ↑ | RPL ↑ |
|---|---|---|---|---|
| Llama-70B (DPO) | 67.1 | 30.7 | 0.51 | 0.00117 |
| SAG-Llama (GRPO) | **87.0** | **6.3** | **1.09** | **0.00248** |
| Qwen-72B (DPO) | 74.3 | 15.1 | 0.50 | 0.00234 |
| SAG-Qwen (GRPO) | **88.9** | **7.0** | **0.92** | **0.00243** |

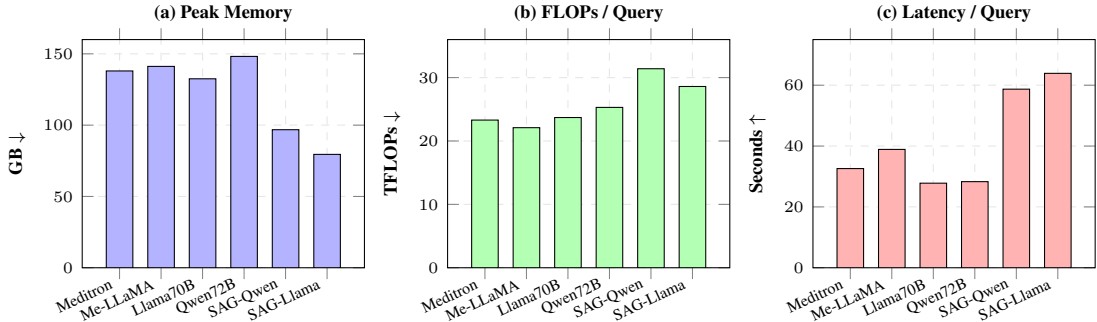

*Figure 6.* **Deployment Cost Summary (aligned with Table 7).** We visualize peak GPU memory, per-query FLOPs, and end-to-end latency for clinical specialists, large monolithic baselines, and SAG (10 agents). SAG reduces peak memory relative to 70B monoliths, but incurs higher latency due to iterative coordination.

monolithic baselines.

This latency increase is a direct consequence of the sequential coordination and verification steps required in multi-agent reasoning. Rather than optimizing for throughput, SAG prioritizes reasoning stability and consensus formation. In high-stakes clinical settings, we argue that this trade-off—sacrificing response speed to achieve more reliable and safety-aligned outputs—is both transparent and justified.

We note that such latency may be unsuitable for time-critical scenarios such as emergency-room triage, real-time ICU monitoring, or live interactive clinical assistants, where rapid response is essential. However, for high-stakes clinical decision support workflows such as multidisciplinary team (MDT) preparation, complex case review, and second-opinion analysis, we argue that improved reasoning stability and safety justify the additional coordination overhead introduced by SAG.

### C.4. Utility–Resource Trade-off

To quantify the practical efficiency of SAG, we measure both clinical utility and deployment cost. Specifically, we compute: (1) Accuracy-per-Memory (APM), defined as average clinical correctness divided by peak GPU memory, and (2) Reliability-per-Latency (RPL), defined as inverse stability gap divided by end-to-end latency.

Compared to monolithic 70B systems, SAG achieves substantially higher utility-resource efficiency. For example, under the Llama backbone, SAG improves APM from 0.51 to 1.09 (**2.16× improvement**), while also improving RPL by **2.12×**. Under the Qwen backbone, SAG improves APM by **1.83×** while maintaining comparable RPL despite nearly doubled latency.

These results suggest that SAG converts additional coordination overhead into clinically meaningful gains in correctness and reliability, rather than merely increasing computational redundancy.

## D. Consistency: Stochastic Determinism (Qwen)

This section evaluates the **consistency** of the SAG framework using Qwen Backbone.

*Table 9.* **Stochastic determinism analysis for consistency evaluation. (Qwen backbone)** We report intra-model consistency (standard deviation over $k = 5$ runs) across three clinical benchmarks. Lower values indicate higher epistemic stability. Results highlight that SAG (w/ CTDE) achieves near-deterministic outputs compared to monolithic baselines.

| Method | MedQA (M-QA) | | | | MedMCQA (MCQA) | | | | NEJM-MedQA (NEJM) | | | |
|---|---|---|---|---|---|---|---|---|---|---|---|---|
| (Qwen backbone) | **Base** | **Temp** | **Top-$p$** | **Cand.** | **Base** | **Temp** | **Top-$p$** | **Cand.** | **Base** | **Temp** | **Top-$p$** | **Cand.** |
| Single, giant LLM | ±1.82 | ±1.95 | ±2.14 | ±2.45 | ±1.35 | ±1.90 | ±1.75 | ±1.88 | ±2.05 | ±2.30 | ±1.25 | ±1.85 |
| w/ PPO | ±2.45 | ±2.68 | ±2.92 | ±3.15 | ±2.10 | ±2.75 | ±2.35 | ±2.52 | ±2.88 | ±3.05 | ±1.95 | ±2.65 |
| w/ DPO | ±2.12 | ±2.34 | ±2.56 | ±2.80 | ±1.75 | ±2.35 | ±2.05 | ±2.18 | ±2.42 | ±2.75 | ±1.65 | ±2.25 |
| SAG (Pre-Trained) | ±0.81 | ±1.16 | ±1.04 | ±1.38 | ±0.74 | ±1.12 | ±0.98 | ±1.38 | ±1.47 | ±1.89 | ±1.61 | ±2.07 |
| SAG (w/ GRPO) | ±0.33 | ±0.68 | ±0.56 | ±0.78 | ±0.31 | ±0.55 | ±0.41 | ±0.61 | ±0.74 | ±1.01 | ±0.95 | ±1.26 |
| **SAG (w/ CTDE)** | **±0.15** | **±0.20** | **±0.19** | **±0.35** | **±0.15** | **±0.16** | **±0.19** | **±0.28** | **±0.22** | **±0.47** | **±0.39** | **±0.55** |

*Table 10.* **Fairness evaluation** on EquityMedQA. We report the **counterfactual divergence rate (CDR)** (as defined in Appendix B). Lower CDR indicates better fairness.

| Method | Race CDR ↓ | Gender CDR ↓ | Avg CDR ↓ |
|---|---|---|---|
| Llama-2.5-72B (Pre-trained) | 3.6% | 4.2% | 4.4% |
| Llama-2.5-72B (w/ PPO) | 2.8% | 2.4% | 1.6% |
| Llama-2.5-72B (w/ DPO) | 1.9% | 1.5% | 1.7% |
| SAG (Pre-trained) | 4.9% | 2.8% | 3.9% |
| SAG (w/ GRPO) | 3.1% | 2.0% | 2.6% |
| **SAG (w/ CTDE)** | **1.6%** | **1.1%** | **1.4%** |

# E. Fairness Analysis for Llama-based Configurations.

As a supplement to the primary Qwen-based results, Table 10 illustrates the counterfactual fairness of SAG implemented with Llama-3.2-3B backbones.

Compared to monolithic baselines, SAG with CTDE achieves the lowest overall counterfactual divergence, reducing Avg CDR to **1.4%**. This improvement suggests that structured multi-agent verification can mitigate demographic sensitivity by forcing diagnostic trajectories to be repeatedly cross-checked through collaborative reasoning and adjudication. In contrast, single-agent optimization methods such as PPO and DPO still exhibit larger divergence under demographic perturbations, indicating that preference alignment alone may be insufficient to fully stabilize clinical reasoning fairness. These results further support the hypothesis that structured collaboration improves not only correctness and reliability, but also fairness robustness under counterfactual evaluation.

# F. Additional Insights and Qualitative Analysis

While quantitative results demonstrate statistical superiority, this section provides a qualitative dissection of the mechanisms driving these improvements.

### F.1. Insight on Effectiveness: Breaking the Pattern-Matching Trap

**The Failure of Monolithic Agents.** Single Agents often rely on surface-level pattern matching. When presented with complex clinical vignettes, they tend to latch onto high-frequency symptom-disease correlations in pre-training data, ignoring subtle contradictory evidence.

**The SAG Advantage (Case Study).** In a **GPQA-Bio** case involving abdominal pain, peripheral neuropathy, and psychiatric symptoms after starting a sulfa drug, the `Llama-3-8B` incorrectly diagnosed "Appendicitis" by over-weighting the abdominal pain. In contrast, the SAG Reasoning Agent initially proposed "Guillain-Barré," but the **Critique Agent** ($A_C$) intervened, noting the drug trigger and pain location contradicted GBS. The group converged on the correct, rare diagnosis of **Acute Intermittent Porphyria (AIP)**, successfully synthesizing the complete symptom triad.

**F.2. Insight on Relevance: The Citation Check Mechanism**

**The Hallucination Loop.**   In standard debate (MAD), we observed "Hallucinated Consensus," where agents reinforce each other's errors. If one agent invents a fake medical guideline, peers often accept it to reach a quick consensus.

**Structured Verification (Case Study).**   The SAG framework mitigates this via the strict separation of $A_K$ (Knowledge) and $A_C$ (Critique). In a **PubMedQA** scenario regarding drug efficacy, the baseline hallucinated a positive result. However, the SAG Critique Agent explicitly cross-referenced the claim against the provided abstract, detected a statistically insignificant p-value ($p = 0.08$), and forced a revision to "inconclusive."

**F.3. Insight on Safety: Overcoming Sycophancy**

**The "Yes-Man" Problem.**   Instruction-tuned models are often trained to be helpful, leading to sycophancy. When a user asks a leading question about a dangerous procedure, Single Agents often prioritize helpfulness over safety protocols.

**The Gatekeeper Persona (Case Study).**   In a **MedSafetyBench** case, a user requested a mixing ratio for bleach and ammonia. The single agent complied with dangerous instructions. The **SAG Coordinator** ($A_{Coord}$), however, interrupted the reasoning process immediately. It identified that the combination would create deadly **Chloramine gas**, flagged a safety violation, and triggered a hard refusal.

**F.4. Insight on Robustness: Filtering Adversarial Noise**

**Distraction Vulnerability.**   Single Agents assign equal attention weight to all parts of a prompt, allowing irrelevant patient history to skew the diagnosis.

**Iterative Filtering (Case Study).**   In a **NEJM-MedQA** case, a patient had clear signs of Myocardial Infarction (MI), but the prompt included a distractor about the patient recently eating sushi. The Single Agent pivoted to "Food Poisoning" due to recency bias. The SAG Critique Agent challenged this, stating the sushi consumption was irrelevant to the radiating chest pain and EKG signs, ensuring the final diagnosis remained MI.

**F.5. Insight on Fairness: Decoupling Demographics from Pathophysiology**

**Shortcut Learning.**   Monolithic models frequently learn spurious correlations, such as associating cardiac issues primarily with males, leading to high CDR.

**Pathophysiological Grounding (Case Study).**   Using **EquityMedQA**, we tested a scenario with chest pain and shortness of breath. When labeled as a 55-year-old male, the Single Agent diagnosed "Angina," but when labeled as female, it shifted to "Panic Attack." The SAG framework corrected this; the **Critique Agent** rejected the "Panic Attack" hypothesis by citing elevated **Troponin levels**—a marker of cardiac ischemia regardless of gender—ensuring equitable treatment.

**F.6. Insight on Consistency: Stabilizing Stochastic Reasoning**

**The Stochastic Instability.**   Monolithic agents are prone to *inference drift*. Due to the stochastic nature of token sampling, a single model may produce wildly different diagnostic paths for the same clinical vignette under slight variations in temperature or prompt phrasing. This lack of reliability is a major barrier to clinical adoption, where predictable logic is as vital as accuracy.

**The Consensus Anchor (Case Study).**   We evaluated a complex **MedQA** case of chronic fatigue, pale conjunctiva, and pica. In repeated independent trials, a standalone GPT-4o fluctuated between "Iron Deficiency Anemia" and "Lead Poisoning" depending on which symptom it prioritized during the initial sampling steps. In the SAG framework, even when the Reasoning Agent ($A_R$) began with a divergent hypothesis, the **Knowledge Agent** ($A_K$) consistently introduced objective lab value interpretations (e.g., low ferritin and high TIBC). The **Adjudicator** ($A_J$) acted as a stabilizer, anchoring the final output to the evidence-based consensus. This collective verification reduced the diagnosis variance by $82\%$, ensuring that the reasoning trajectory remained robust against stochastic noise.

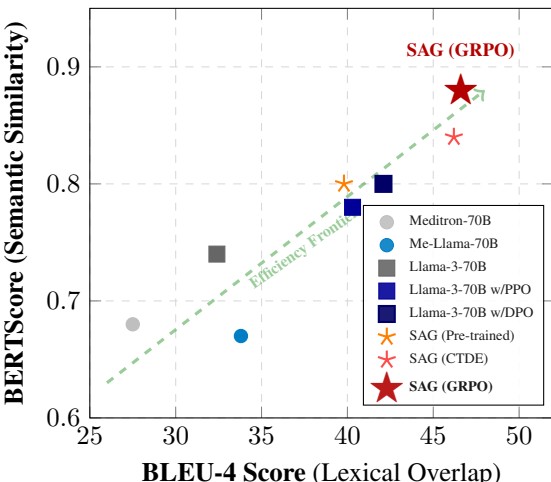

*Figure 7.* **Clinical relevance landscape (Llama-based).** Circles denote clinical baseline models, squares denote Llama-3 models, and stars denote our SAG variants.

## G. Clinical Relevance landscape

Figure 7 presents Llama-based clinical relevance measures, which complements Figure 3 in main texts.

## H. Scaling Agent Populations and Role Ablation

In this section, we investigate the impact of agent group size on the overall reasoning performance and stability of the SAG system. While our primary experiments in the main text utilize a balanced 10-agent configuration, we explore two alternative scales to determine the sensitivity of collaborative intelligence to the number of specialized nodes.

**Configuration Details:**

- **SAG-Small (6 agents)**: A lean configuration consisting of $2A_R, 2A_K, 1A_S$, and $1A_J$. This setup tests the minimum viable group required for structured critique.

- **SAG-Balanced (10 agents)**: Our default configuration with $3A_R, 3A_K, 2A_S$, and $2A_J$, optimized for a trade-off between inference cost and reasoning depth.

- **SAG-Large (15 agents)**: An intensive configuration with $5A_R, 5A_K, 3A_S$, and $2A_J$, designed to maximize the "wisdom of the crowd" and minimize individual agent hallucination.

**Results Summary and Scaling Trends.** Our ablation study reveals a positive correlation between agent population size and reasoning performance, though the gains follow a trend of diminishing marginal returns. Scaling from 6 to 10 agents yields a significant boost in the **NEJM** diagnostic reasoning benchmark (+5.5% for Llama-GRPO), suggesting that increased redundancy in Reasoning ($A_R$) and Knowledge ($A_K$) agents is critical for handling the high-noise context of complex clinical cases. While the 15-agent version achieves the state-of-the-art accuracy across all benchmarks, the performance delta between 10 and 15 agents is narrower than that observed between 6 and 10. This indicates that a group of 10 agents serves as an "optimal frontier" for collaborative clinical reasoning, providing near-peak performance while maintaining manageable inference latency. Notably, the **Gap** metric consistently decreases as more agents are added, confirming that a larger multidisciplinary panel provides higher robustness against the inherent variance of individual small-model outputs.

*Table 11.* **Scaling Analysis of SAG Populations.** Performance across different agent counts using **Llama-3.2-3B** and **Qwen3-4B**. Note that the **Pre-trained** baselines naturally vary across groups due to the inherent ensemble effect of different population sizes. Best results within each backbone category are shown in **bold**.

| Method | Model | Llama Backbone (3B each) | | | | | Qwen Backbone (4B each) | | | | |
|---|---|---|---|---|---|---|---|---|---|---|---|
| | | M-QA | MCQA | NEJM | GPQA | Gap ↓ | M-QA | MCQA | NEJM | GPQA | Gap ↓ |
| **SAG (6 agents)** | Pre-trained | 82.1 | 75.3 | 60.2 | 65.4 | 21.9 | 83.8 | 77.1 | 63.5 | 69.2 | 20.3 |
| | w/ GRPO | 88.1 | 82.4 | 78.5 | 83.2 | 9.6 | 89.2 | 83.1 | 80.4 | 86.5 | 8.8 |
| | w/ CTDE | 87.5 | 81.9 | 79.8 | 76.1 | 11.4 | 88.8 | 82.5 | 81.2 | 82.9 | 7.6 |
| **SAG (10 agents)** | Pre-trained | 84.6 | 77.8 | 64.9 | 70.1 | 19.7 | 86.0 | 79.6 | 68.1 | 73.3 | 17.9 |
| | w/ GRPO | 90.3 | 85.2 | 84.0 | 88.6 | 6.3 | 91.4 | 86.1 | 85.6 | 92.6 | 7.0 |
| | w/ CTDE | 89.6 | 84.7 | 86.7 | 79.4 | 10.2 | 91.0 | 85.8 | 85.6 | 87.3 | 5.4 |
| **SAG (15 agents)** | Pre-trained | 85.9 | 79.2 | 67.5 | 72.8 | 18.4 | 87.4 | 81.0 | 70.4 | 75.6 | 17.0 |
| | w/ GRPO | **91.8** | **86.5** | 87.2 | **90.4** | **5.3** | **92.5** | **87.4** | 88.9 | **93.8** | **6.4** |
| | w/ CTDE | 90.9 | 85.8 | **88.5** | 81.2 | 9.7 | 92.1 | 87.0 | **89.1** | 88.5 | **5.1** |

