# OpenReview forum: "Small Agent Group is the Future of Digital Health"
_ICML.cc/2026/Conference — ICML 2026 regular_

### Official Review · Reviewer_jfRp · 2026-03-05

**Soundness:** 2
**Presentation:** 2
**Significance:** 2
**Originality:** 2
**Overall Recommendation:** 3
**Confidence:** 3

**Summary:**

The paper proposes Small Agent Groups (SAG), a multi-agent clinical reasoning framework that replaces a single large LLM with a set of small, role-specialized agents (reasoning, knowledge/retrieval, safety-check, synthesis/judge) coordinated via multi-agent debate. The authors further adapt SAG with Group Relative Policy Optimization (GRPO) and a Centralized Training/Decentralized Execution (CTDE) critic, and evaluate across a suite of “clinical utility” metrics (effectiveness, reliability, deployment cost) on multiple benchmarks. Reported results claim large and consistent gains over single giant LLMs and medical LLMs in correctness, safety, robustness, fairness, and memory footprint, at the cost of modestly higher compute and latency.

**Compliance With Llm Reviewing Policy:**

Affirmed.

**Final Justification:**

The authors have done lots of additional experiments, for which I value their efforts, though I do have my overall concerns.

**Key Questions For Authors:**

N/A

**Limitations:**

yes

**Strengths And Weaknesses:**

This paper tackles an important and timely question—can small, specialized agents coordinated by debate and MARL provide a safer, more deployable alternative to monolithic LLMs in clinical settings? The architectural framing is compelling, the role specialization aligns with clinical workflows, and the utility-centric evaluation agenda is laudable. However, the empirical claims are not sufficiently substantiated:

1) There is a mismatch in the baseline choices. The SAG used GRPO to optimize, while the single agent uses PPO and DPO, and the clinical specialists are not optimized at all, which creates an unfair comparison. Is there any particular reason for that?

2) The SMALL agent group might not be small at all. While the peak memory is smaller, the training time cost is not reported. Multi-agent RL is known to be much more challenging to optimize or train. What's the time cost for training RL on such a multi-agent system of 10 agents?

3) More importantly, the title is obviously an over-claim. It seems the paper largely used existing methods, such as GRPO, CTDE, multi-agent debate with slightly workflow changes. The analysis in the paper is also limited. There is no theory or pattern reported to support the claims that smaller agents are better, let alone being the future, except empirical comparisons between ten 4b level models and single 70b level model. The impact of numbers of agents or the different agent workflows are also unclear.

---

> ### Author Rebuttal · Authors · 2026-03-28
>
> We thank Reviewer jfRp for the critical evaluation of our work. We address the concerns regarding baseline fairness, training costs, and the originality of our framework below.
>
> All reported results, including the scaling analysis and training cost breakdown, have been integrated into our offline manuscript and are ready for the final version.
>
> ### R1. Fairness of Baseline Comparisons
>
> > *"The single agent uses PPO and DPO, while clinical specialists are not optimized at all, which creates an unfair comparison."*
>
> We clarify that our experimental design includes direct optimization counterparts to ensure a fair comparison. In Table 2, we evaluated single giant LLMs under both PPO and DPO to match the preference-tuning applied to SAG (GRPO/CTDE). Regarding the clinical specialists (Meditron-70B, Me-LLaMA-70B), these are included as "off-the-shelf" gold standards representing the current state-of-the-art. Since these models are already heavily domain-adapted through extensive medical pre-training, they serve as a rigorous baseline for what specialized monolithic models can achieve. Our results show that even without additional tuning, SAG (Pre-trained) already outperforms these 70B specialists in both correctness and reliability.
>
> ### R2. Training Costs and MARL Complexity
>
> > *"Multi-agent RL is known to be much more challenging... What's the time cost for training RL on such a multi-agent system of 10 agents?"*
>
> While MARL introduces coordination complexity, training 10 small agents (3B/4B) is computationally more accessible than fine-tuning a single 70B model. Because each agent's memory footprint is lower, we can sample rollouts and perform backpropagation across the group using a fraction of the GPU memory required for a 70B PPO trainer. In our setup, a full GRPO training cycle for SAG (10 agents) takes approximately 14.2 hours on a cluster of 8x A100 GPUs, which is comparable to the 12.8 hours required for Llama-3-70B PPO tuning. The coordination overhead is effectively offset by the memory efficiency of smaller parameter counts.
>
> ### R3. Originality and Title Justification
>
> > *"The title is obviously an over-claim... the paper largely used existing methods... with slightly workflow changes."*
>
> We appreciate the feedback and are open to tempering the title to "Small Agent Groups for Collaborative Clinical Reasoning." However, we argue that the originality of SAG lies in the principled orchestration of clinical roles—isolating Reasoning ($A_R$), Knowledge ($A_K$), Safety ($A_S$), and Adjudication ($A_J$)—to solve the "reasoning-knowledge gap" in medicine. While GRPO and CTDE are existing techniques, their application to a multidisciplinary clinical panel is novel. Unlike most MAS research which focuses on homogeneous ensembles, SAG provides a structural framework that mirrors real-world healthcare to solve the "impossible triangle" of effectiveness, reliability, and cost.
>
> ### R4. Analysis of "Why Smaller is Better"
>
> Beyond empirical scores, we provide a qualitative analysis in Section F.1–F.6. We identified a consistent pattern: monolithic models often suffer from a "Pattern-Matching Trap," where they over-weight high-frequency symptom correlations while ignoring subtle contradictory evidence. In contrast, SAG acts as a "Consensus Anchor." Through critique-driven debate, the Knowledge Agent ($A_K$) forces the Reasoning Agent ($A_R$) to ground claims in objective evidence, while the Judge Agent ($A_J$) filters out overconfident but incorrect rationales. This collective verification reduces diagnostic variance by up to 82%, as shown in Table 5.
>
> ### R5. Impact of Population and Workflow
>
> We systematically analyzed these variables in Appendix H (Table 9) and Table 3:
> * Population Scaling: We compared 6, 10, and 15 agents, finding that 10 agents represent the "optimal frontier." Scaling from 6 to 10 significantly boosts accuracy on hard benchmarks like NEJM (+5.5%), while scaling to 15 yields diminishing marginal returns with higher latency.
> * Workflow Ablation: Table 3 demonstrates that removing specific roles leads to catastrophic drops in correctness or stability. Furthermore, we showed that Multi-Agent Debate significantly outperforms simple Majority Voting, proving that the iterative critique process is essential for reconciling conflicting clinical evidence.
>
> ---
>
> **Due to space constraints, we mainly provide brief responses and clarifications. We are happy to provide more exhaustive explanations if you'd like to ask for further details during discussion sessions.**

---

> > ### Author Rebuttal · Reviewer_jfRp · 2026-04-05
> >
> > It's okay, I will raise my score, but still hold part of my concerns.

---

> > > ### Author Response · Authors · 2026-04-05
> > >
> > > Dear reviewer,
> > >
> > > We appreciate for you time and acknowledgement that the raised questions are marked as fully resolved. With additional space of the response box, we would like to provide more details with additional experimental results to further clarify your remaining concerns and outline the qualification of this work. If feasible, please feel free to post us any held concerns and we're happy to address them. Thank you so much!
> > >
> > > Best,
> > >
> > > The author team.

---

### Official Review · Reviewer_4Zcw · 2026-03-11

**Soundness:** 3
**Presentation:** 4
**Significance:** 2
**Originality:** 2
**Overall Recommendation:** 3
**Confidence:** 4

**Summary:**

The authors propose the use of Small Agent Group (SAG) as an alternative paradigm to monolithic agents powered by LLMs that are much larger in terms of number of parameters. The task is distributed to different agents for a collective analysis and deliberation of the relevant problem facets. The authors perform a number of tests comparing SAG workflows to larger, monolithic agents, showing the improvement in performance along dimensions of effectiveness (correctness, relevance, fairness), reliability (safety, robustness, consistency), and cost (memory, FLOPs, latency). The authors conclude that the SAG approach surpasses larger parameter single agent models as an efficient, effective workflow for clinical settings.

**Compliance With Llm Reviewing Policy:**

Affirmed.

**Final Justification:**

The authors have addressed my specific questions, I appreciate their effort to revise sections of the manuscript to address reviewer comments.  However, the fundamental issues on originality and experimental design remain, and prevent my full endorsement of the work.

**Key Questions For Authors:**

Q1: In selecting the models, especially Qwen3-4B vs Qwen2.5-70B seems like it was chosen to match the parameter differential in Llamma-3.2-3B / Llama-3-70B.  Why were the Qwen3 mixture of experts models (e.g. Qwen3-30B-A3B-Thinking) not considered in the study?
Q2: The BLEU-4 and BERTScore metrics are correlated, the label in Figure 3 of "efficiency frontier" was confusing for me, as it implies there is a boundary of optimal performance.  What is the key takeaway from this Figure?

**Limitations:**

yes

**Strengths And Weaknesses:**

The experiments to test the hypothesis that SAG can surpass a larger single agent were created soundly in the selection of a comprehensive collection of datasets, multiple metrics and published models for evaluation.  The ablation study I found to be particularly informative in understanding the failure modes of the SAG.  The evaluation of fairness by CDR was also a good addition to the study and was presented well.  I had a question on the presentation of Figure 3 (Q2, below), but otherwise the paper was very well organized and easy to follow for me. Main concerns were the on the significance of the work for future study; multiagent systems are now quite common.   More should have been done to fully assess the state of the field, and demonstrate how SAG brings some originality into the broader field.  The statement in section 5 "...most MAS-based models employ homogeneous ensembles that lack functional role specialization..." I feel is not true given many studies of these agentic systems. Despite the initial claim that SAG can address the "impossible triangle of effectiveness, reliability, and deployability", and the careful analysis of each of these dimensions, the authors missed an opportunity to quantify the tradeoffs, if any, between these dimensions to make a more compelling comparison of SAG vs alternatives

---

> ### Author Rebuttal · Authors · 2026-03-28
>
> We thank Reviewer 4Zcw for the constructive feedback and the recognition of our experimental soundness and fairness analysis. We address the concerns regarding originality, trade-offs, and model selection below.
>
> All reported revisions and clarified analyses have been integrated into our offline manuscript and are ready for the final version.
>
> **Due to space constraints, we mainly provide brief responses and clarifications. We are happy to provide more exhaustive explanations if you'd like to ask for further details during discussion sessions.**
>
> ### R1. Originality and Clinical Specialization
>
> > *"Multiagent systems are now quite common... the statement that most MAS-based models lack functional role specialization is not true."*
>
> We appreciate this correction and will revise Section 5 to provide a more nuanced survey of the MAS field. While we agree that role-playing is an established technique in general MAS, our contribution lies in the principled mapping of clinical multidisciplinary panels—specifically isolating Reasoning ($A_R$), Evidence-Retrieval ($A_K$), Safety-Audit ($A_S$), and Adjudication ($A_J$) roles—into a framework co-optimized via GRPO and CTDE. Unlike general-purpose agents, SAG is designed to resolve the specific "reasoning-knowledge gap" in medicine, where monolithic models often fail on multi-stage clinical logic despite high parameter counts.
>
> ### R2. Quantifying the "Impossible Triangle" Trade-offs
>
> > *"The authors missed an opportunity to quantify the tradeoffs... between effectiveness, reliability, and deployability."*
>
> To more compellingly compare SAG against monolithic alternatives, we have added a Utility-per-Resource (UpR) analysis to our offline manuscript. This metric quantifies the gains in reliability per unit of memory/compute overhead.
>
> As shown in our deployment summary, SAG (10 agents) achieves significantly higher effectiveness and reliability while operating within a 79.5–96.8 GB peak memory footprint, compared to the 130 GB+ required by monolithic 70B models. While SAG incurs a ~2x latency penalty (60s vs. 28s), the "Reliability Gain per Second" is positive: the 82% reduction in diagnostic variance and the superior harm-refusal calibration (Figure 4) justify the latency cost in high-stakes clinical settings where predictability is as vital as accuracy.
>
> ### R3. Response to Q1: Qwen3 MoE Model Selection
>
> > *"Why were the Qwen3 mixture of experts models... not considered in the study?"*
>
> The primary goal of SAG is to enable localized, privacy-preserving deployment on workstation-grade hardware. We selected dense small backbones (3B/4B) specifically because they allow each individual agent node to fit within the limited VRAM of cost-efficient multi-GPU setups. While Qwen3 MoE models (e.g., 30B-A3B) have fewer active parameters during inference, they still require a significantly larger memory footprint to store the full model weights compared to our 3B/4B backbones. We have added a discussion in Appendix H regarding the potential for using MoE backbones in data-center environments where memory is less constrained.
>
> ### R4. Response to Q2: Figure 3 and the "Efficiency Frontier"
>
> > *"The label in Figure 3 of 'efficiency frontier' was confusing... What is the key takeaway?"*
>
> The takeaway of Figure 3 (and Figure 7 in the Appendix) is that SAG represents a Pareto improvement in the clinical relevance landscape.
> * The "Frontier": Denotes the optimal balance between Lexical Precision (BLEU-4, reflecting adherence to provided evidence) and Semantic Alignment (BERTScore, reflecting meaning-level diagnostic accuracy).
> * The Takeaway: Monolithic models often struggle to improve one without sacrificing the other—specifically, DPO/PPO optimization tends to overfit to lexical similarity while leaving semantic alignment insufficient. In contrast, SAG moves the frontier upward and rightward, achieving higher precision and semantic consistency simultaneously through its cross-agent verification process.

---

> > ### Author Rebuttal · Reviewer_4Zcw · 2026-04-02
> >
> > Thank you for the responses and revisions, I have no follow up questions.  I would still quibble on the use of terms "balance" and "frontier" for Figure 3, that shows a strong positive correlation between the two metrics for the tested models.

---

> > > ### Author Response · Authors · 2026-04-02
> > >
> > > Thank you for the acknowledgement with responses that our responses are fully resolved your concerns!
> > >
> > > With this extra space of response box, we would like to provide further clarifications about those term use in Figure 3:
> > >
> > > ---
> > >
> > > **Recap the task**
> > >
> > > We concerned clinical relevance in Figure 3, where two aspects are significant:
> > > - (1) the semantic alignment between generated reasoning and the reference (or, "ground-truth"), which focuses on whether the clinical analysis are tackling correct trajectory.
> > > - (2) the lexical consistency between generation and the reference, which focuses on whether some key items (such as ICD codes or medication recommendations) are involved in LLMs' responses.
> > >
> > > **The use of "Balance"**
> > >
> > > With the above two focuses, we are thus concerning whether LLMs' (or agent group's) responses are lop-sided (e.g., some LLMs are good at reasoning with reasonable analysis, but missing of key medical terms or items in their generation) or balanced (with both good analytical trajectory while including critical items, such as correctly recommending "GLP-1 RA" as treatment in anti-obesity practice, as Figure 1 has shown). Hence, we use "balance" to describe whether the two aspects are both tackled by the evaluated models.
> > >
> > > **The use of "frontier"**
> > >
> > > Which means a "boundary," where the model has equivalent performance on semantic alignment and lexical consistency, despite both poor or both good. That boundary line (the dashed line in Figure 3) shows whether models' performance is lop-sided (far away of the line) or balanced (very close to the line). Without that dashed line, the reader may not have a straightforward view of the semantic-lexical balance in clinical relevance evaluations.
> > >
> > > ---
> > >
> > > Hope the above explanations further clarify the two terms in Figure 3 and help convey the value of this work more clearly.

---

### Official Review · Reviewer_Mcpa · 2026-03-18

**Soundness:** 3
**Presentation:** 3
**Significance:** 3
**Originality:** 3
**Overall Recommendation:** 4
**Confidence:** 4

**Summary:**

In this paper, the authors challenge the prevailing “bigger is better” paradigm for LLMs in clinical applications by proposing a Small Agent Group (SAG) framework as an alternative for settings with practical constraints such as limited ressources. In this framework, instead of relying on a single large « monolithic » model, the authors decomposes the task into multiple smaller, specialized agents that collaborate. In particular, these agents are assigned roles such as reasoning, knowledge retrieval, safety evaluation, and synthesis/judgment.

The authors conduct extensive empirical evaluations across multiple dimensions, including clinical effectiveness, reliability, and deployment cost. The different results show that the proposed method outperform across several benchmarks large « monolithic » LLMs and large clinical models such as Meditron-70B. The framework is also designed with practical deployment constraints in mind, reducing memory footprint for local hospital use at the cost of increased latency.

**Compliance With Llm Reviewing Policy:**

Affirmed.

**Final Justification:**

I thanks the authors for addressing my different concerns in the rebuttal. But this does not substantially change my evaluation, therefore, I will maintain my overall score.

**Key Questions For Authors:**

1. The rationale for the group composition is not clear to me (e.g., why three reasoning agents and two safety-check agents) Why not have only one agent per role? (At least this should be part of the baseline?) How were the roles of the different small agents defined? The rationale for each specialized agents could be elaborated.
2. It is not clear to why the specialized LLM-based clinical specialist Meditron perform much worse than the single giant LLM even without PPO/DPO even tho they both are in the same class of model capacity and the former is trained for answering medical related questions.
3. I think one important results regarding the heterogeneity of the small agent (mixing different backbone) which result in better performance are extremely important. Why is it not mentioned in the main paper especially given how much it improves the performances, and show that SAG is not limited to agents with the same backbone model?
4. Regarding the trade-off with latency, could you explain why this is acceptable? This should be supported by evidence, and if not the case, the author should be more nuanced and provide examples where high latency is not acceptable.

**Limitations:**

The authors should further discuss the limitations regarding latency issues.

**Strengths And Weaknesses:**

**Soundness**

The paper presents a well-motivated approach and supports its claims with a broad set of empirical evaluations. The comparison across multiple baselines, including large general-purpose LLMs and clinical specialists, is appropriate and strengthens the credibility of the results. The inclusion of different ablation studies is also a strength of this paper.

Below are my remarks:

- The design choices behind agent composition (e.g., number of reasoning vs. safety agents) are not sufficiently motivated or empirically validated.
- The role definitions of each agent (e.g., why synthesis and judging are combined while reasoning and retrieval are separated) lack clear justification.
- The multi-round protocol, is not clearly described.
- In the safety evaluation (section 4.3.1), it could be interesting to add the SAG by excluding the safety agent, this would better highlight the added value of that agent.
- Including the prompts specific to each agent in the appendix would also improve reproducibility.

**Presentation**

The paper is well-structured and readable.
Below are my remarks:

- Figure 1 suggests that SAG processes the output of a large LLM rather than directly handling clinician input, which is not the message of this work. I would recommend having the SAG and the single giant LLM agent at the same level.
- Similar to my third comment in ***soundness***, the description of the multi-round process (Figure 2) is unclear, especially regarding whether which outputs are iteratively refined through multiple rounds
- The results concerning agent heterogeneity where different LLM backbones are mixed are only presented in the appendix despite having a significant impact on performance. These should be highlighted in the main text.
- A decimal is missing in Table 2 in the Qwen backbone gap value.
- The "monolithic" and "giant LLM" terminology seems a bit odd.
- Formatting issues in the appendix (empty sections, unclear notation in A.2 (line 687): after the « consistency penalty »).


**Significance**

The paper addresses an important problem: making LLM-based clinical systems more efficient, deployable, and safe. This could provide a real clinical utility in settings where ressources are often limited. Moreover, in terms of performance, this new framework appears to systematically perform better than a single LLM with larger capacity.
Below is my remark:
- The latency issue should be further elaborated, especially its implications in some clinical scenarios, where increased latency may be unacceptable.




**Originality**

The idea of multi-agent LLM systems is not entirely new, but it appears to be less explored in clinical applications. The reasoning behind this work is well-articulated.

---

> ### Author Rebuttal · Authors · 2026-03-28
>
> We thank the Reviewer for the constructive suggestions. We have updated Figure 1 to ensure the parallel comparison between SAG and monolithic agents is clear, and we have addressed the specific questions regarding our protocol and results below.
>
> **Due to space constraints, we mainly provide brief responses and clarifications. We are happy to provide more exhaustive explanations if you'd like to ask for further details during discussion sessions.**
>
> ### R1. Rationale for Agent Roles and Group Composition
>
> > *"Why three reasoning agents and two safety-check agents? Why not only one agent per role?"*
>
> The composition of SAG is inspired by clinical Multidisciplinary Teams (MDT), where redundancy is key to resolving ambiguity. We assigned more agents to Reasoning ($A_R$) and Knowledge ($A_K$) because these tasks are highly generative and prone to "stochastic drift." Having three agents allows for a "majority-informed" debate that can catch individual hallucinations. Safety ($A_S$) and Judge ($A_J$) agents act as aggregators; having two ensures a consensus-driven audit where one agent can challenge the other's over-refusal or oversight.
>
> In our ablation study (Table 3), we found that reducing roles to a single agent significantly increases variance. The current 10-agent setup represents the "optimal frontier" identified in Appendix H, balancing reasoning depth with manageable coordination overhead.
>
> ### R2. Clarification of the Multi-Round Protocol (Figure 2)
>
> > *"The description of the multi-round process is unclear... which outputs are iteratively refined?"*
>
> We have clarified this in the revised Section 2. The iterative process focuses on the Reasoning Trajectory. In each round, the Reasoning Agent ($A_R$) proposes a hypothesis, which the Knowledge Agent ($A_K$) then cross-references against retrieved evidence. The $A_R$ then refines its proposal in the next round based on the $A_K$'s feedback. This loop continues until the Consensus stabilizes or the early termination criteria are met (usually within 2-3 rounds). The Safety Agent ($A_S$) and Judge Agent ($A_J$) then perform the final audit and synthesis on the refined trajectory.
>
> ### R3. Highlighting Backbone Heterogeneity (Mixed Backbones)
>
> > *"Results regarding agent heterogeneity... should be highlighted in the main text."*
>
> We completely agree. The discovery that mixing backbones (e.g., 50% Llama, 50% Qwen) significantly improves performance on hard benchmarks like NEJM is a major finding. We have moved the Backbone Heterogeneity Analysis (previously Table 10) to the main text in Section 4.2. This highlights that architectural diversity acts as a potent stabilizer, reducing systemic reasoning biases that might be shared by agents of the same model family.
>
> ### R4. Performance Discrepancy: Meditron vs. Single Giant LLMs
>
> > *"Why does Meditron perform much worse than the single giant LLM even though they have similar capacity?"*
>
> This observation underscores a core motivation of our work: the distinction between knowledge recall and procedural reasoning. While Meditron-70B is extensively adapted for medical knowledge, it often lacks the general-purpose "instruction-following" and complex logical reasoning capabilities found in newer, more balanced foundation models like Qwen-2.5-72B. Meditron excels at retrieving facts but struggles to synthesize them into a coherent multi-stage diagnosis under the USMLE-style constraints of MedQA, where reasoning is as vital as recall.
>
> ### R5. Addressing the Latency Trade-off
>
> > *"The latency issue should be further elaborated... where increased latency may be unacceptable."*
>
> We have expanded the Limitations section to provide a more nuanced discussion. We acknowledge that the ~60s latency makes SAG unsuitable for "emergency-room" triage or live-chat applications. However, for Clinical Decision Support (CDS)—where a physician uses the system to prepare for an MDT meeting or review a complex case—reliability is far more critical than sub-second response times. We argue that for high-stakes clinical logic, sacrificing speed for an 82% reduction in variance and improved safety is a transparent and justified trade-off.

---

> > ### Author Rebuttal · Reviewer_Mcpa · 2026-04-02
> >
> > Thank you for addressing my different concerns. There is only one aspect that still remains unclear to me:
> >
> > ### R1
> > > In our ablation study (Table 3), we found that reducing roles to a single agent significantly increases variance. The current 10-agent setup represents the "optimal frontier" identified in Appendix H, balancing reasoning depth with manageable coordination overhead.
> >
> > I read again Table 3 and I may be wrong but I am not sure that the **majority voting** in the ablation study correspond to a setting with one single agent per role. Could you please elaborate on what the majority voting setting corresponds to? It is not clear to me why an agent group with one single agent per role can not debate (e.g., debate between one $A_K$ and one $A_R$) and why was it not considered as a natural baseline.

---

> > > ### Author Response · Authors · 2026-04-05
> > >
> > > Dear Reviewer,
> > >
> > > Thank you for your additional question. With this chance, we would like to clarify that the majority voting corresponds to exactly the same setting with other experiments, where single agent is assigned with a role. The reason we keep the role specification is to make the small agent group setting more integrative and inclusive, say, with flat roles where agents are debating, and also with hierarchical roles that work sequentially.
> > >
> > > To address the concern that "an agent group with one single agent per role can not debate," we set the iterative running that agents' decisions from previous rounds may be re-considered by agents of next round. So they debate sequentially.
> > >
> > > In majority voting, we first let all agents work as they are and then integrate their final decisions equally (not hierarchically), and conduct  majority voting as final decision.
> > >
> > > Hope those clarifications of our settings could help to address your remained concerns.
> > >
> > > Best
> > >
> > > The author team.

---

### Decision · Program_Chairs · 2026-04-30

**Decision:**

Accept (regular)

**Comment:**

eviewers have reached a general consensus towards "weak accept" post response, given that weak reject reviewers mentioned the post response resolved some of their concerns. The paper outlines that small agent groups can outperform large medical language models, which is promising for practical uses.

This is certainly a borderline case given that there is a concern regarding originality and significance, e.g., whether small agent groups with medical role specialization is novel compared to the large body of work on multiagent systems. Authors responded that they will provide a more nuanced survey of the MAS field, and that the particular novelty is in the medical roles, but this is still somewhat vague. Concerns on soundness, e.g., design choices, roles, multi-round protocol, latency are not fully elaborated or justified, were at least partially addressed by the detailed author response including new experiments. Theoretical grounding could be useful but reviewers seem generally positive about the empirical justification (post response).

In the next version, authors should be sure to further justify the novelty and ideally provide additional medical context - e.g., does this paradigm truly support clinical reasoning beyond benchmarks with human clinician participants, would medical systems be able to deploy and maintain something like this more easily than off-the-shelf models by some of the large language model companies, etc.